# Cell cycle regulation of the psoriasis associated gene *CCHCR1* by transcription factor E2F1

**Yick Hin Ling**[1¤]*, **Yingying Chen**[2], **Kwok Nam Leung**[2], **King Ming Chan**[2], **W. K. Liu**[1]

**1** School of Biomedical Sciences, Faculty of Medicine, The Chinese University of Hong Kong, Shatin, New Territories, Hong Kong, **2** School of Life Sciences, Faculty of Science, The Chinese University of Hong Kong, Shatin, New Territories, Hong Kong

¤ Current address: Department of Biology, Johns Hopkins University, Baltimore, Maryland, United States of America

* yhinling@gmail.com

**Data Availability Statement:** All relevant data are within the paper and its Supporting information files.

**Funding:** The author(s) received no specific funding for this work.

## Abstract

The coiled-coil alpha-helical rod protein 1 (CCHCR1) was first identified as a candidate gene in psoriasis and has lately been found to be associated with a wide range of clinical conditions including COVID-19. CCHCR1 is located within P-bodies and centrosomes, but its exact role in these two subcellular structures and its transcriptional control remain largely unknown. Here, we showed that *CCHCR1* shares a bidirectional promoter with its neighboring gene, *TCF19*. This bidirectional promoter is activated by the G1/S-regulatory transcription factor E2F1, and both genes are co-induced during the G1/S transition of the cell cycle. A luciferase reporter assay suggests that the short intergenic sequence, only 287 bp in length, is sufficient for the G1/S induction of both genes, but the expression of *CCHCR1* is further enhanced by the presence of exon 1 from both *TCF19* and *CCHCR1*. This research uncovers the transcriptional regulation of the *CCHCR1* gene, offering new perspectives on its function. These findings contribute to the broader understanding of diseases associated with CCHCR1 and may serve as a foundational benchmark for future research in these vital medical fields.

## Introduction

*CCHCR1* (coiled-coil alpha-helical rod protein 1) is a candidate gene for psoriasis, a skin condition affecting 1% to 2% of the population [1]. It has been implicated in various cellular processes including keratinocyte proliferation and differentiation [2, 3], steroidogenesis [4, 5], myogenic differentiation [6], and cytoskeleton organization [7]. Studies from our lab have uncovered CCHCR1 as a novel protein component of P-bodies and centrosome [8], suggesting its potential dual roles in regulating RNA metabolism and microtubules organization, respectively. This finding is further supported by RNA sequencing that highlights its role in related pathways [9], and knockdown of CCHCR1 caused centriole duplication defects and multipolar spindle formation [10]. Notably, the *CCHCR1* gene has recently been linked to alopecia areata [11], type-2 diabetes [12], Takayasu arteritis [13], and COVID-19 susceptibility [14], emphasizing the importance of understanding its cellular function and transcription regulation.

**Competing interests:** NO authors have competing interests.

Psoriatic skin is characterized by hyper-proliferation and abnormal differentiation of keratinocytes [15]. Early studies attempted to establish a correlation between the expression of CCHCR1 and cell proliferation, but the relationship is complex. Positive correlations are found with insulin and estrogen, which promote keratinocyte growth and up-regulate *CCHCR1* [5]. The immunosuppressant cyclosporine A, which inhibits keratinocyte proliferation, results in its down-regulation [5]. The inactivation of the Rb pathway in neuronal cells triggers cell cycle re-entry and *CCHCR1* up-regulation [16]. On the contrary, negative correlations are observed with serum starvation followed by re-feeding of human HaCaT keratinocytes, which leads to cell cycle re-entry but down-regulates *CCHCR1*. The topoisomerase inhibitor camptothecin (CPT), which causes DNA damage, growth arrest, and cell apoptosis [17–20], up-regulates *CCHCR1* [21]. Intriguingly, overexpression of CCHCR1 reduces keratinocyte proliferation. Given the varying experimental conditions and cell types examined, which have resulted in diverse and sometimes contradictory findings, it is essential to understand the fundamental mechanisms of the transcriptional regulation of *CCHCR1* to reconcile these discrepancies. In this study, we demonstrated that *CCHCR1* shares a bidirectional promoter with its neighboring gene, *TCF19* (transcription factor 19). The activity of this bidirectional promoter is controlled by E2F1, a transcription factor involved in the G1/S cell cycle transition. In line with this, both *CCHCR1* and *TCF19* are co-induced during the G1/S transition. Our findings underscore the intricate cell cycle regulation of *CCHCR1*, paving the way for a deeper understanding of its role in various cellular processes and associated diseases.

## Results

### E2F1 regulates *CCHCR1-TCF19* bidirectional promoter

The PSORS1 (Psoriasis Susceptibility 1) locus, identified as one of the most significant genetic determinants for psoriasis susceptibility, is located on the short arm of chromosome 6 (6p21.3) within the major histocompatibility complex (MHC), spanning approximately 200 kb (22, 23). Within the PSORS1 locus, we found that the *CCHCR1* gene is oriented head-to-head with its neighboring gene, *TCF19*, and their transcriptional start sites are only 287 bp apart (Fig 1A and 1B). The close proximity between the two genes suggests the possibility of co-regulation. Typically, gene pairs with a bidirectional promoter are separated by less than 1000 bp [22]. The nucleotide sequence preceding the start codon of *CCHCR1* and *TCF19* reveals typical characteristics of a classical bidirectional promoter [22]: the region is TATA-less and contains a CpG island that overlaps their first exons (Fig 1B); moreover, multiple putative transcription factor binding sites that are over-represented in the bidirectional promoter [23], such as GABP, MYC, E2F1, E2F4, NFY, and YY1 are found in the intergenic region between these two genes (S1 and S2 Tables). Notably, using MatInspector transcription factor binding site matrices [24], we found that the intergenic region is highly enriched with E2F transcription factor binding motifs (Fig 1B and S2 Table), a family of transcription factors that are instrumental in controlling the expression of genes that play a role in the G1/S transition and DNA synthesis during S phase [25]. Analysis of the microarray database [26] revealed a positive correlation between *CCHCR1* expression and genes associated with the G1/S transition or S phase of the cell cycle, including *E2F1* and *TCF19* (S3 Table). Furthermore, a positive correlation between *CCHCR1* and *TCF19* expression across various cell types suggests potential co-regulation (S3 Table).

The E2F transcription factor family consists of 8 members: E2F1-3 are activators, and E2F4-8 are repressors. To address the role of E2F in the expression of *CCHCR1* and *TCF19*, we measured the mRNA levels of *CCHCR1* and *TCF19* after ectopic overexpression of E2F1, 2, or 3 in HeLa cells. Among these three E2F activators, only E2F1 overexpression induces *CCHCR1*

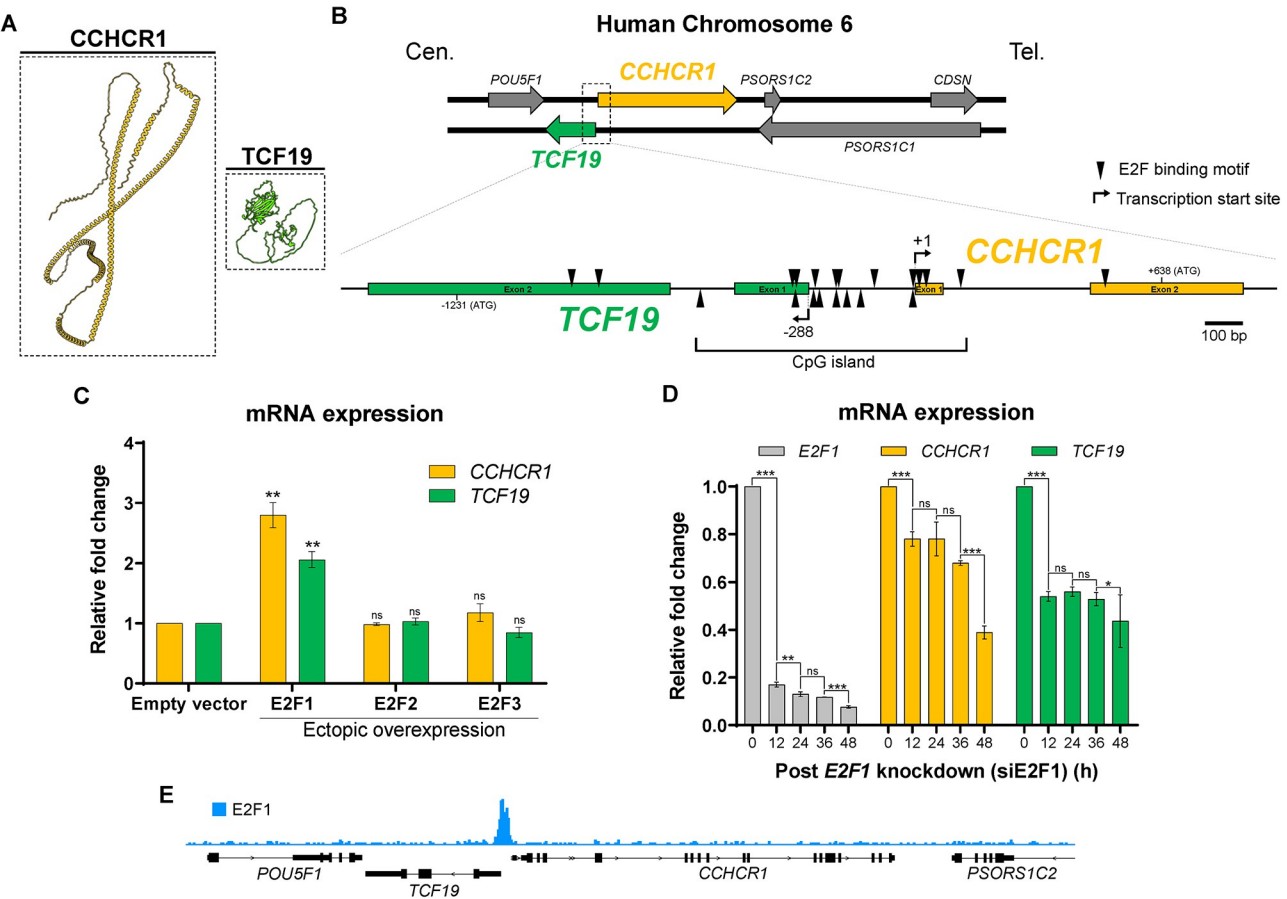

**Fig 1. E2F1 induces the expression of *CCHCR1* and *TCF19*.** (A) AlphaFold predicted structure of CCHCR1 and TCF19 proteins. (B) Genome arrangement of the *CCHCR1* and *TCF19* on human chromosome 6. Cen.: Centromeric side, Tel.: Telomeric side. Putative E2F binding motifs (arrowhead) and CpG island shown. Numbers refer to the position related to the transcription start site (+1) of *CCHCR1*. (C) qPCR for the expression of *CCHCR1* and *TCF19* in HeLa cells transfected with E2F activators (E2F1, E2F2 or E2F3). (D) Down-regulation of *CCHCR1* and *TCF19* mRNA by *E2F1* mRNA knockdown (siE2F1). (E) E2F1 shows high occupancy in the intergenic region of *CCHCR1-TCF19*. ChIP-seq data retrieved from ChIP-Atlas [SRX150563] [27].

and *TCF19* (Fig 1C). In addition, siRNA-mediated knockdown of *E2F1* resulted in a time-dependent decrease in the expression levels of *CCHCR1* and *TCF19*. 48 hours post-siRNA treatment, *E2F1* mRNA levels significantly decreased, corresponding to the lowest observed expression levels of both *CCHCR1* and *TCF19* (Fig 1D). This correlation highlights the potential role of E2F1 as a transcriptional activator for these two genes. Notably, E2F1 shows high occupancy in the *CCHCR1-TCF19* intergenic region (Fig 1E), as indicated by chromatin immunoprecipitation (ChIP) in HeLa cells from ChIP-Atlas [27]. Therefore, E2F1 is functionally linked to the expression of *CCHCR1* and *TCF19*.

## *CCHCR1* and *TCF19* are induced in the G1/S transition

Since E2F1 regulates genes in the G1/S transition and S phase [25], we explored the cell cycle expression of *CCHCR1* and *TCF19*. HeLa cells were synchronized at the G1/S transition by double thymidine block, and then released to allow cell cycle progression (Fig 2A and 2B). The expression patterns of *CCHCR1* and *TCF19* are closely correlated (Fig 2C). Their expression peaks at the G1/S transition (0 h), remains high during the S phase (2 h), decreases during the

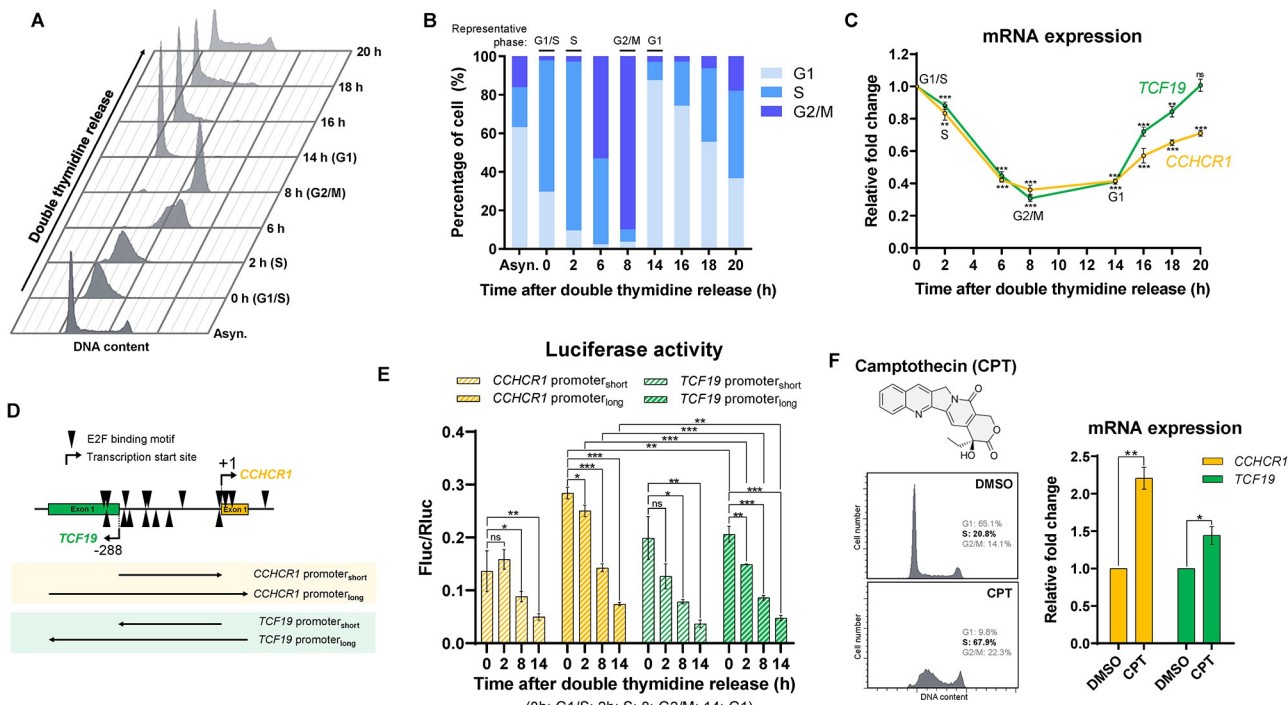

**Fig 2. *CCHCR1* and *TCF19* are induced in the G1/S transition.** (A) DNA content and (B) percentage of cells in different cell cycle phases synchronized by double thymidine block and release. Asyn.: Asynchronous. (C) Relative expression of *CCHCR1* and *TCF19* in different cell cycle phases. (D) DNA fragments of *CCHCR1-TCF19* bidirectional promoter for the dual luciferase assay. (E) Normalized luciferase activity (Fluc/Rluc) for *CCHCR1-TCF19* bidirectional promoter fragments. (F) DNA content (right) and relative expression of *CCHCR1* and *TCF19* (left) after 2 μM CPT treatment for 24 h.

transition to the G2/M phase (8 h), and continues to be low in the G1 phase (14 h) (Fig 2C). This cell cycle-dependent expression pattern is consistent with the notion that the two genes are co-regulated by E2F1. Although cells begin to lose synchrony particularly from 16 h after release, complicating the clear dissection of the G1/S transition in subsequent cell cycles (Fig 2A and 2B), we still observed a coordinated increase in the expression of *CCHCR1* and *TCF19* when cells accumulated in the S phase (Fig 2C).

To verify the cell cycle-dependent activity of the *CCHCR1-TCF19* bidirectional promoter, we cloned promoter fragments into firefly luciferase (Fluc) reporter constructs and performed a dual luciferase assay to monitor their activities throughout the cell cycle. The short promoter fragments (287 bp) contain the intergenic sequence in either forward or reverse orientations (Fig 2D). The long promoter fragments (556 bp) additionally include exon 1 of both genes (Fig 2D). These promoter fragments show high luciferase activities during the G1/S transition (0 h) and S phase (2 h) (Fig 2E). Specifically, the long fragment in the forward orientation for *CCHCR1* demonstrates higher activity across all cell cycle phases, compared to the corresponding fragment in the reverse orientation for *TCF19* (Fig 2E). Overall, we showed that the activity of the *CCHCR1-TCF19* bidirectional promoter is induced in the G1/S transition and maintained at a high level in S phase.

Recognizing the cell cycle regulation of *CCHCR1* and *TCF19* prompts us to further examine the effect of camptothecin (CPT) on their expression. CPT, a topoisomerase I inhibitor that induces cell cycle arrest at the G2 and S phases [17–20], has been reported to up-regulate the expression of *CCHCR1* [21] and *E2F1* [28, 29]. We observed prominent S phase arrest after a

24-hour treatment with 2 μM CPT (Fig 2F; left), and importantly, both *CCHCR1* and *TCF19* are up-regulated (Fig 2F; right), suggesting that CPT-induced S-phase arrest could explain the up-regulation of both genes. This finding underscores the importance of examining the cell cycle status when interpreting the effects of drugs and experimental conditions on *CCHCR1* gene expression.

## Discussion

The *CCHCR1* gene, well known as a candidate gene for psoriasis, has received ample attention in clinical research in the last decade. In addition to psoriasis [30–32], it has potential associations with various diseases such as alopecia areata [11], type-2 diabetes [12], Takayasu arteritis [13], and COVID-19 [14]. Early genetic association studies showed controversial results for *CCHCR1* as the psoriasis susceptibility gene, possibly due to the small sample sizes, different populations and different statistical analyses being used [30–32]. As psoriasis is a complex multifactorial disease [33], it is also likely that one risk allele cannot fully explain the pathogenesis. Recognizing the limitation of clinical association studies, we aim to provide clarity on the biological regulation of *CCHCR1*. In this study, we found that *CCHCR1* transcription is activated by the G1/S-regulatory transcription factor E2F1, within a bidirectional promoter shared with *TCF19*. We hypothesized that this configuration allows synchronized induction of both genes during the G1/S transition of the cell cycle.

Human genome sequencing revealed an abundance (>10%) of divergently transcribed gene pairs whose transcription start sites are separated by less than 1000 bp [34]. A shared part of the intergenic fragment initiates and regulates transcription in both directions [34]. These gene pairs are highly co-expressed [35] and may function in the same biological pathway such as DNA repair, cell cycle process, chromatin modification, metabolic process, and cell proliferation and differentiation [34, 36]. The cellular function of *TCF19* is still vague. It was predicted as a transcription factor due to the presence of a putative trans-activating domain [37]. Mechanistically, TCF19 was shown to recognize H3K4me3 [38] to control the expression of glucose-responsive genes [38, 39]. Similar to CCHCR1, TCF19 has been associated with diseases, including diabetes [40–42], HBV-related chronic hepatitis B, cirrhosis, hepatocellular carcinoma [43], non-small cell lung cancer [44], squamous cell carcinoma of the head and neck (SCCHN) [45] and colorectal cancer [46]. It is currently not known whether CCHCR1 and TCF19 act on the same cellular pathway. However, one thing to note is that both genes are linked to diabetes. Interestingly, the head-to-head arrangement of *CCHCR1* and *TCF19* is conserved in mice, with intergenic region separated by 291 bp. Further investigation into the conserved genomic arrangement of *CCHCR1-TCF19* could uncover potential shared pathways in disease processes, enhancing our understanding of their functional interplay and providing insights for targeted therapeutic strategies.

We and others have identified CCHCR1 as a novel protein component in P-bodies and centrosomes [7, 8, 10, 47]. In this study, we demonstrated that *CCHCR1* is highly expressed during the G1/S transition and S phase of the cell cycle, potentially linking its activity to crucial cell cycle events within these subcellular structures. P-bodies are phase-separated condensates in the cytoplasm responsible for RNA processing. The size and number of P-bodies change dynamically throughout the cell cycle, increasing in the S phase and disassembling upon mitotic entry [48]. Inside the P-bodies, CCHCR1 interacts with the mRNA decapping protein EDC4 [8, 47]. However, its exact role in mRNA processing remains unclear. In the centrosome, CCHCR1 controls cytoskeletal organization [7], and its knockdown resulted in centriole duplication defects and multipolar spindle formation [10]. Notably, centrosome duplication and centriole replication occur just before, or at the onset of the S phase [49], coinciding with

the induction time of *CCHCR1* (Fig 2C). Further investigations into how CCHCR1 impacts the dynamics of both P-bodies and centrosomes throughout the cell cycle could provide vital insights into its cellular roles.

## Materials and methods

### *In silico* analysis of the transcriptional regulation of *CCHCR1*

The intergenic region of *CCHCR1-TCF19* was analyzed by EMBOSS Newcpgreport [50] for the presence of CpG islands, and by MatInspector [24] for putative transcription factor binding sites. Genes positively correlated with *CCHCR1* expression were identified using Genevestigator [26].

### Quantitative PCR (qPCR)

Total RNA from HeLa cells was isolated using TRIzol® Reagent (Life Technologies, USA) according to the manufacturer's instructions. Extracted RNAs were treated with DNase to remove genomic DNA using a TURBO DNA-free™ Kit (Life Technologies, USA), and were reverse transcribed to first-strand cDNAs with random primers using PrimeScript First Strand cDNA Synthesis Kit (Takara, Japan). Quantitative PCR (qPCR) was performed using Fast SYBR Green Master Mix (Life Technologies, USA) in a Step One Plus Real-time quantitative PCR system (Life Technologies, USA), with ROX as the reference dye. A final dissociation stage was performed after the thermocycles to verify the specificity of the PCR primers. All amplifications were done in triplicate. The relative expression of gene transcripts was normalized to *RPL13A* expression, and gene expression levels were calculated using the Pfaffl method [51]. The primer pairs used are listed in S4 Table.

### Overexpression of E2F proteins

To overexpress E2F1, 2, or 3, the coding sequences of the corresponding E2Fs were cloned from cDNA of HeLa cells into pEGFP-N1 vectors, which contain a strong CMV promoter. The plasmids were transiently transfected into HeLa cells using Fugene HD transfection reagent (Roche, Germany) for 48 hours to express the corresponding E2F-EGFP fusion proteins. Successful transfection was verified by the GFP expression of the cells.

### siRNA transfection

HeLa cells were seeded in a 6-well plate overnight before being transfected with 2.5 μM of siRNAs using Lipofectamine RNAiMAX transfection reagent (Invitrogen, USA) according to the manufacturer's instructions. *CCHCR1*, *TCF19*, and *E2F1* mRNA were depleted using predesigned Silencer® Select siRNAs (Ambion, Life Technologies, USA). The negative control siRNA (siCTL) was Silencer® Select Negative Control No. 1 siRNA (#4390843, Ambion, Life Technologies, USA). We observed no significant change of *CCHCR1* and *TCF19* mRNA level after 48 hours post-transfection of siCTL, while strong depletion of *CCHCR1* and *TCF19* mRNA was observed when treated with their specific siRNAs (S1 Fig).

### Cell cycle synchronization

Cells were synchronized at the G1/S transition by a double thymidine block. Briefly, HeLa cells at 25–30% confluency were incubated with 2 mM thymidine in complete DMEM medium for 18 hours, released into the thymidine-free complete medium for 9 hours, and incubated again with 2 mM thymidine in complete DMEM medium for another 17 hours. The cells were released from the block by washing with PBS and incubated with the complete DMEM

medium. Fig 2A–2C and 2E illustrate representative cell cycle phases at different time points following the double thymidine release, as determined by the distribution of DNA content (Fig 2B). At time 0 hours, immediately following the release of the double thymidine block, cells are defined as being at the G1/S transition. Other representative phases are assigned when a particular stage surpasses 85% of the population, i.e. S phase at 2 hours, G2/M phase at 8 hours, and G1 phase at 14 hours.

## Flow cytometry

Cells were trypsinized, fixed in 70% ethanol, stained with propidium iodide, and analyzed by flow cytometry using LSRFortessa Cell Analyzer (BD Biosciences, USA) for DNA content.

## Dual luciferase assay

Promoter fragments containing the *CCHCR1-TCF19* intergenic region were cloned from HeLa cell genomic DNA and inserted into the Firefly luciferase (Fluc) reporter vector pGL4.17 (Promega, USA), in either forward or reverse orientation. Firefly luciferase activity was measured to monitor the transcription activities of various *CCHCR1-TCF19* intergenic fragments at different phases of the cell cycle. In brief, HeLa cells ($1 \times 10^6$) were seeded in 24-well plates and incubated for 24 hours before transfection. Double transfections were performed using 200 ng total DNA containing 100 ng of Fluc reporter vector and 100 ng of Renilla luciferase (Rluc) vector (pRL-TKl; Promega). After 4 hours of transfection, cells were synchronized by double thymidine treatment and harvested at different time points after release from the block. Cells were lysed and luciferase activity was determined using a Dual-Luciferase Reporter Assay System (Promega) with a GloMax-96 Microplate Luminometer (Promega) as previously described [52]. To calculate the luciferase activity, the Firefly luciferase (Fluc) signal was normalized to the Renilla luciferase (Rluc) signal to account for transfection efficiency variations. Then, the Fluc/Rluc ratio of the empty vector control (pGL4.17) was subtracted from that of each promoter fragment.

## Statistical analysis

All experiments were done in triplicate and data are presented as the mean value and standard deviation. Statistical analysis was performed with GraphPad Prism software version 7.0 (GraphPad Software, USA). Unpaired t-test was used to determine the significance of the data. Statistical significance levels are denoted as: * $P \leq 0.05$, ** $P \leq 0.01$ and *** $P \leq 0.001$; ns, not significant.

## Supporting information

**S1 Table. Putative transcription factor binding sites in the *CCHCR1-TCF19* bidirectional promoter.**
(PDF)

**S2 Table. Putative E2F binding motif in the *CCHCR1-TCF19* bidirectional promoter.**
(PDF)

**S3 Table. *CCHCR1* co-expressed genes.**
(PDF)

**S4 Table. Primers for quantitative PCR.**
(PDF)

**S1 Fig. siRNAs control experiment.**
(PDF)

**S1 File. References for supporting information.**
(PDF)

# Acknowledgments

We thank Prof. Zhixiu Lin of the School of Chinese Medicine for technical assistance; Scarlet Cho, Gan Ling and Cho Ling for helpful discussions.

# Author Contributions

**Conceptualization:** Yick Hin Ling.

**Data curation:** Yick Hin Ling, Yingying Chen, W. K. Liu.

**Formal analysis:** Yick Hin Ling, Yingying Chen, W. K. Liu.

**Funding acquisition:** W. K. Liu.

**Investigation:** Yick Hin Ling.

**Methodology:** Yick Hin Ling.

**Resources:** Kwok Nam Leung, King Ming Chan, W. K. Liu.

**Supervision:** W. K. Liu.

**Writing – original draft:** Yick Hin Ling.

**Writing – review & editing:** Yick Hin Ling.

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
