## [Decision Letter · Decision Letter 0]

23 Oct 2023

PONE-D-23-25477Cell cycle regulation of the psoriasis associated gene CCHCR1 by transcription factor E2F1PLOS ONE

Dear Dr. Ling,

Thank you for submitting your manuscript to PLOS ONE. After careful consideration, we feel that it has merit but does not fully meet PLOS ONE’s publication criteria as it currently stands. Therefore, we invite you to submit a revised version of the manuscript that addresses the points raised during the review process.

 Please see the comments from two reviewers below and in the attachment. Please note that some of the recommendations may be outside the scope of the study, and we invite you to rebut these concerns should you wish.Please ensure that the code and data is shared upon resubmission as per PLOS ONE policy, and indicate where these can be found in the Data availability statemenPlease submit your revised manuscript by Dec 02 2023 11:59PM. If you will need more time than this to complete your revisions, please reply to this message or contact the journal office at plosone@plos.org. Please include the following items when submitting your revised manuscript:A rebuttal letter that responds to each point raised by the academic editor and reviewer(s). You should upload this letter as a separate file labeled 'Response to Reviewers'.A marked-up copy of your manuscript that highlights changes made to the original version. You should upload this as a separate file labeled 'Revised Manuscript with Track Changes'.An unmarked version of your revised paper without tracked changes. You should upload this as a separate file labeled 'Manuscript'.If applicable, we recommend that you deposit your laboratory protocols in protocols.io to enhance the reproducibility of your results. Protocols.io assigns your protocol its own identifier (DOI) so that it can be cited independently in the future. For instructions see: https://journals.plos.org/plosone/s/submission-guidelines#loc-laboratory-protocols. Additionally, PLOS ONE offers an option for publishing peer-reviewed Lab Protocol articles, which describe protocols hosted on protocols.io. Read more information on sharing protocols at https://plos.org/protocols?utm_medium=editorial-email&utm_source=authorletters&utm_campaign=protocols.

We look forward to receiving your revised manuscript.

Kind regards,

Hanna Landenmark

Staff Editor

PLOS ONE

Journal Requirements:

Reviewers' comments:

Reviewer's Responses to Questions

**Comments to the Author**

1. Is the manuscript technically sound, and do the data support the conclusions?

Reviewer #1: Yes

Reviewer #2: Partly

2. Has the statistical analysis been performed appropriately and rigorously? 

Reviewer #1: Yes

Reviewer #2: Yes

3. Have the authors made all data underlying the findings in their manuscript fully available?

Reviewer #1: Yes

Reviewer #2: Yes

4. Is the manuscript presented in an intelligible fashion and written in standard English?

Reviewer #1: Yes

Reviewer #2: No

5. Review Comments to the Author

Reviewer #1: The study is technically well executed and controlled and provides essential insights into the cell cycle regulation of CCHCR1. The data fit well with the topics of PLOS ONE. Therefore, I think that, once solved some minor issue, this manuscript is suitable for publication on PLOS ONE. See the attach

Reviewer #2: This article has a novel approach and demonstrates that CCHCR1 shares a bidirectional promoter with its neighboring gene TCF19. The activity of this bidirectional promoter is regulated by the transcription factor E2F1 involved in G1/S cell cycle transition, however there are still several issues.1. How to find the TCF19 gene when studying the CCHCR1 gene?2. What experimental method was used to measure the highly enriched E2F transcription factor binding motif in the CCHCR1-TCF19 gene region?3. In Figure 1D, the expression of CCHCR1 and TCF19 genes at different time points should be compared, and the reasons for the significant differences should be explained in the discussion?4. In each figure of Figure 2, the representative cell cycle should be indicated on the graph, and which time point represents the G1/S phase should be clearly stated in the article, and the basis for staging should be explained?5. This article only conducted research on cell regulation, without exploring the cellular and systemic effects caused by genes. It is recommended to increase research in this area to highlight the exact role of CCHCR1-TCF19 gene in psoriasis.

6. PLOS authors have the option to publish the peer review history of their article (what does this mean?). If published, this will include your full peer review and any attached files.

Reviewer #1: No

Reviewer #2: No

---

## [Author Response · Author response to Decision Letter 0]

27 Oct 2023

Dear Editor and Reviewers,

We extend our sincere thanks for the comprehensive review of our manuscript and the constructive feedback provided. We have addressed each point raised by the referees, conducting additional statistical analyses and revising the manuscript accordingly. Our point-by-point responses are highlighted in blue in the 'Response to Reviewers' file, and changes made to the manuscript text can be found in the 'Revised Manuscript with Track Changes' file.

Reviewer #1: The study is technically well executed and controlled and provides essential insights into the cell cycle regulation of CCHCR1. The data fit well with the topics of PLOS ONE. Therefore, I think that, once solved some minor issue, this manuscript is suitable for publication on PLOS ONE. See the attach

In this manuscript the authors investigated the genomic arrangement of CCHCR1 gene. They demonstrate that CCHCR1 shares a bidirectional promoter with its neighboring gene TCF19 and that this bidirectional promoter is activated by the G1/S-regulatory transcription factor E2F1, and both genes are co-induced during the G1/S transition of the cell cycle.

The study is technically well executed and controlled and provides essential insights into the cell cycle regulation of CCHCR1. The data fit well with the topics of PLOS ONE.

Therefore, I think that, once solved some minor issue, this manuscript is suitable for publication on PLOS ONE

Response: We would like to thank Reviewer #1 for the positive reception of our manuscript. We have addressed the referee's comments as detailed below.

Minor point:

-Some concerns with the data showing the expression peak of TCF19 at the G1/S transition. In Figure 2C the increased expression at G1/S transition does not seems clear to me. TCF19 levels at time 0h after double thymidine release are very similar to 20h level. The authors may want to strengthen this claim with additional experiments or add a comment about the expression level of TCF19 at 20h post release. 

Response: We have revised our manuscript to clarify that E2F1 controls gene expression during both the G1/S transition and the S phase, as noted in line 88. In Figure 2C, time 0 h corresponds to the G1/S transition. However, from 16-20 h, the cells begin to lose synchrony, resulting in a mixed population primarily in the S and G1/S phases. At these time points, there is no predominant accumulation of cells in the G1/S phase compared to the 0 h timepoint. Consequently, we believe that a direct comparison between 0 h and 20 h might be complex. The increase in the expression levels of TCF19 (and CCHCR1) from 16-20 h is interpreted as an indication that an increasing number of cells are progressing into the S phase. We have provided a more comprehensive discussion on this topic from lines 91 to 97 on page 4 of the manuscript.

-In order to make the statement, “Specifically, the long fragment in the forward orientation for CCHCR1 demonstrates higher activity across all cell cycle phases, unlike the corresponding fragment in the reverse orientation for TCF19”, a statistic analysis between the luciferase activity of the long promoter fragment in the forward orientation for CCHCR1 and the luciferase activity of the corresponding promoter fragment in the reverse orientation for TCF19 needs to be performed. Only if it is significant that statement might be assessed. Otherwise, it is possible a statement of the observation.

Response: We have conducted a statistical analysis to compare the luciferase activity of the long promoter fragment in the forward orientation for CCHCR1 with the luciferase activity of the corresponding promoter fragment in the reverse orientation for TCF19, as depicted in the revised Figure 2E. The results demonstrate a statistically significant difference, providing support for our statement.

Reviewer #2: This article has a novel approach and demonstrates that CCHCR1 shares a bidirectional promoter with its neighboring gene TCF19. The activity of this bidirectional promoter is regulated by the transcription factor E2F1 involved in G1/S cell cycle transition, however there are still several issues.

Response: We sincerely thank Reviewer #2 for the insightful review and constructive comments on our manuscript. We have carefully addressed each of the concerns raised to enhance the clarity and quality of our work.

1. How to find the TCF19 gene when studying the CCHCR1 gene?

Response: In our exploration of the PSORS1 locus on chromosome 6, a crucial region for psoriasis susceptibility, we identified the unique head-to-head orientation of the CCHCR1 gene with the TCF19 gene. The proximity of their transcriptional start sites intrigued us and led to further investigation of their co-transcriptional regulation. We have provided a more comprehensive discussion of this topic from lines 47 to 51 of the manuscript.

2. What experimental method was used to measure the highly enriched E2F transcription factor binding motif in the CCHCR1-TCF19 gene region?

Response: We used MatInspector, a bioinformatic tool that allows the identification of putative E2F motifs through a comprehensive library of weight matrices. This information has been added at line 58 of the manuscript.

3. In Figure 1D, the expression of CCHCR1 and TCF19 genes at different time points should be compared, and the reasons for the significant differences should be explained in the discussion?

Response: We have revised Figure 1D to compare the expression of CCHCR1 and TCF19 at different time points. To avoid cluttering the plot with multiple statistical significances, we selectively showed the statistical significance for comparing each time point with the one just before it. We believe this revision more clearly indicates a time-dependent decrease in mRNA levels.

mRNA knockdown by siRNA is time-dependent and maximal knockdown is often seen 24 - 48 hours after transfection. In the Results section, we clarified that the expressions of the CCHCR1 and TCF19 genes demonstrate a time-dependent reduction following E2F1 siRNA treatment. The pronounced decrease in the expression levels of both genes, especially at 48 hours, correlates with a significant reduction in E2F1 mRNA levels, underscoring E2F1's potential role as a transcriptional activator for CCHCR1 and TCF19. This topic is discussed in detail from lines 78 to 82 on page 4 of the manuscript.

4. In each figure of Figure 2, the representative cell cycle should be indicated on the graph, and which time point represents the G1/S phase should be clearly stated in the article, and the basis for staging should be explained?

Response: Thank you for highlighting this critical detail. We have updated Figure 2 to better illustrate the representative cell cycle phases. At time 0 hours, immediately following the release of the double thymidine block, cells are defined as being at the G1/S transition. Other representative phases, such as S phase (2 hours), G2/M phase (8 hours), and G1 phase (14 hours), are assigned when a particular stage is predominant in over 85% of the cell populations, as illustrated in Figure 2B. We have added a detailed explanation of the staging process to the Materials and Methods section, from lines 207 to 212 on page 9 of the manuscript. Additionally, the revised manuscript now specifies that time 0 h represents the G1/S phase.

5. This article only conducted research on cell regulation, without exploring the cellular and systemic effects caused by genes. It is recommended to increase research in this area to highlight the exact role of CCHCR1-TCF19 gene in psoriasis.

Response: We appreciate the recommendation from the referee. This study is primarily focused on the transcriptional regulation of CCHCR1 and TCF19. While the exploration of the broader cellular and systemic impacts of these genes in psoriasis is indeed fundamental, it falls beyond the scope of our current work. However, this is a significant aspect that we intend to consider in future research.

---

## [Decision Letter · Decision Letter 1]

7 Nov 2023

Cell cycle regulation of the psoriasis associated gene CCHCR1 by transcription factor E2F1

PONE-D-23-25477R1

Dear Dr. Ling,

We’re pleased to inform you that your manuscript has been judged scientifically suitable for publication and will be formally accepted for publication once it meets all outstanding technical requirements.

Kind regards,

Claude Prigent

Academic Editor

PLOS ONE

Additional Editor Comments (optional):

Reviewers' comments:

Reviewer's Responses to Questions

**Comments to the Author**

1. If the authors have adequately addressed your comments raised in a previous round of review and you feel that this manuscript is now acceptable for publication, you may indicate that here to bypass the “Comments to the Author” section, enter your conflict of interest statement in the “Confidential to Editor” section, and submit your "Accept" recommendation.

Reviewer #1: All comments have been addressed

Reviewer #2: All comments have been addressed

2. Is the manuscript technically sound, and do the data support the conclusions?

Reviewer #1: Yes

Reviewer #2: Yes

3. Has the statistical analysis been performed appropriately and rigorously? 

Reviewer #1: Yes

Reviewer #2: Yes

4. Have the authors made all data underlying the findings in their manuscript fully available?

Reviewer #1: Yes

Reviewer #2: Yes

5. Is the manuscript presented in an intelligible fashion and written in standard English?

Reviewer #1: Yes

Reviewer #2: Yes

6. Review Comments to the Author

Reviewer #1: (No Response)

Reviewer #2: I think the author has provided clear responses to the reviewer's questions, with a more detailed description. Based on the original description, the article has been made clearer, and I believe it is acceptable to accept this article.

7. PLOS authors have the option to publish the peer review history of their article (what does this mean?). If published, this will include your full peer review and any attached files.

Reviewer #1: No

Reviewer #2: No

---

## [Editor Report · Acceptance letter]

12 Dec 2023

PONE-D-23-25477R1 

Cell cycle regulation of the psoriasis associated gene *CCHCR1* by transcription factor E2F1 

Dear Dr. Ling:

I'm pleased to inform you that your manuscript has been deemed suitable for publication in PLOS ONE. Congratulations! Your manuscript is now with our production department. 

Kind regards, 

on behalf of

Dr. Claude Prigent 

Academic Editor

PLOS ONE